# Overview of the Side-Effects of FDA- and/or EMA-Approved Targeted Therapies for the Treatment of Hematological Malignancies

**DOI:** 10.3390/jcm9092903

**Published:** 2020-09-08

**Authors:** Catalin Constantinescu, Sergiu Pasca, Alina-Andreea Zimta, Tiberiu Tat, Ioana Rus, Patric Teodorescu, Sabina Iluta, Alina Tanase, Anca Colita, Olafur Sigurjonsson, Hermann Einsele, Ciprian Tomuleasa

**Affiliations:** 1Intensive Care Unit, Ion Chiricuta Clinical Cancer Center, 400010 Cluj Napoca, Romania; constantinescu.catalin@ymail.com (C.C.); dr_tibi@yahoo.com (T.T.); 2Department of Hematology, Iuliu Hatieganu University of Medicine and Pharmacy, 400004 Cluj Napoca, Romania; pasca.sergiu123@gmail.com (S.P.); patric_te@yahoo.com (P.T.); iluta.sabina@yahoo.com (S.I.); 3Department of Anesthesia and Intensive Care, Iuliu Hatieganu University of Medicine and Pharmacy, 400000 Cluj Napoca, Romania; 4Medfuture Research Center for Advanced Medicine, Iuliu Hatieganu University of Medicine and Pharmacy, 400000 Cluj Napoca, Romania; zimta.alina.andreea@gmail.com; 5Department of Hematology, Ion Chiricuta Clinical Cancer Center, 400014 Cluj Napoca, Romania; codruta_21@yahoo.com; 6Department of Stem Cell Transplantation, Fundeni Clinical Institute, 925200 Bucharest, Romania; alinadanielatanase@yahoo.com (A.T.); ancacolita@yahoo.com (A.C.); 7The Blood Bank, Landspitali–the National University Hospital of Iceland, Snorrabraut 60, 101 Reykjavik, Iceland; oes@landspitali.is; 8Department of Internal Medicine II, University Hospital Würzburg, 97080 Würzburg, Germany; Einsele_H@uw.de

**Keywords:** novel therapies, life-threatening side-effects, hematological malignancies

## Abstract

In the last decade there has been tremendous effort in offering better therapeutic management strategies to patients with hematologic malignancies. These efforts have ranged from biological to clinical approaches and resulted in the rapid development of new approaches. The main “problem” that comes with the high influx of newly approved drugs, which not only influences hematologists that frequently work with these drugs but also affects other healthcare professionals that work with hematologists in patient management, including intensive care unit (ICU) physicians, is they have to keep up within their specialty and, in addition, with the side-effects that can occur when encountering hematology-specific therapies. Nonetheless, there are few people that have an in-depth understanding of a specialty outside theirs. Thus, this manuscript offers an overview of the most common side-effects caused by therapies used in hematology nowadays, or that are currently being investigated in clinical trials, with the purpose to serve as an aid to other specialties. Nevertheless, because of the high amount of information on this subject, each chapter will offer an overview of the side-effects of a drug class with each reference of the section being intended as further reading.

## 1. Introduction

The last decade has brought forward tremendous effort in offering better therapeutic management strategies to patients with hematologic malignancies. These efforts have taken either translational or clinical approaches and resulted in the rapid development of new approaches that are more tailored to the disease and the patient [1]. These range from the development of chimeric antigen receptor T-cells (CAR-T cells) to small inhibitory molecules and different varieties of antibodies. Nevertheless, this rapid development of a wide variety of targets is accompanied by a myriad of side-effects, most of which are interdisciplinary and involve clinical scenarios that are challenging both for hematologists and intensive care unit (ICU) physicians. In treating a patient diagnosed with a hematological malignancy, hematologists and oncologists collaborate with other medical specialties to provide a better outcome, and this often includes the departments of cardiology, pneumology or nephrology. Thus, all clinicians have to be up-to-date with the latest progress in immunotherapy and stem cell transplantation. To meet this endeavor, this manuscript offers an overview of the side-effects of novel therapies for all malignancies in order to improve the management of patients under these treatments.

## 2. Specific Targets in Hematology

Figure 1 shows the typical targets in the main hematological malignancies. Based on this structure, we further present each individual drug for each of these targets and highlight the target-specific side-effects.

Historically, one of the most important targets in malignant hematology is the fusion protein BCR-ABL in chronic myeloid leukemia (CML). The discovery that *BCR-ABL* can be efficiently targeted has greatly improved the survival of CML patients [2]. BCR-ABL inhibitors are classified into three generations, and each of the individual drugs present specific side-effects. Imatinib is the first *BCR-ABL* inhibitor for CML, and it is the first generation of this class of drugs to which other members of the class are frequently compared. Imatinib has a safe side-effect profile, with most patients tolerating it well. Some common side-effects that can appear in this case are represented by nausea, vomiting, muscle and joint pain, headache and diarrhea [3]. The second generation of BCR-ABL inhibitors is represented by nilotinib, dasatinib and bosutinib. It should be mentioned that there is a certain tendency in Europe to use nilotinib and in the United States to use dasatinib. Each of these three has a characteristic side-effect that is worse and more frequent when compared to imatinib. Nilotinib is more frequently associated with cardiovascular events, whereas dasatinib is associated with pleural effusion and bosutinib with gastrointestinal conditions, especially diarrhea. The third generation is represented by ponatinib, which is also the most potent and can be efficient even for CML patients where resistance develops to first- and second-generation drugs. Nonetheless, the most marked side-effect of this drug is represented by cardiovascular events that are worse and more frequent than in the case of nilotinib [4].

Generally, drugs that match the patient’s comorbidities with their side-effects should be avoided. Quite frequently, imatinib will be the first choice when treating CML and will be changed to a second-generation *BCR-ABL* inhibitor if the disease progresses or if the patient presents intolerance to imatinib [4].

One of the targets for a varied repertoire of drugs, which is expressed on the surface of multiple hematological malignancies, is CD20. For anti-CD20 targets, most side-effects are target-related, as anti-CD20 therapy determines B-cell depletion, with most of these patients requiring immunoglobulin substitution [5]. Nonetheless, most cases receiving substitution do not report an increase in infections with encapsulated bacteria, showing the efficacy of this countermeasure. Similarly, some therapies target CD19, which presents a similar mechanism of action as in the case of anti-CD20 therapy. These drugs also target B-cells and require immunoglobulin substitution [6]. Interestingly, when targeting the B-cell maturation antigen (BCMA), no important reduction in immunoglobulins is reported when compared to the case of anti-CD19 or anti-CD20 therapies, even though this antigen is highly expressed on plasma cells. The main side-effects of anti-BCMA therapy are represented by reactions to the drug itself, most commonly infusion-related reactions or allergic reactions, in the case of antibodies, and cytokine release syndrome (CRS) in the case of anti-BCMA CAR-T cells [7].

Another important target for plasma cells is CD38. Targeting CD38 in multiple myeloma generally has mild side-effects, and the infusion-related reaction is the most prominent. Still, patients receiving the anti-CD38 drug daratumumab require a special transfusion, as further described [8]. Brentuximab vedotin, an anti-CD30 therapy, is an excellent therapeutic strategy for classic Hodgkin and peripheral T-cell lymphomas, and it is used as a single-agent monotherapy that bridges the transplant for T-cell lymphomas or is used in combination with ABVD (adriamycin, bleomycin, vinblastine, dacarbazine) chemotherapy for stage IV Hodgkin lymphomas [9]. Nonetheless, in addition to the side-effects associated with the monoclonal antibody-based structure of brentuximab vedotin, CD30 is a marker of activated T-cells, thus potentially leading to an impairment of these cells. Peripheral neuropathy is a side-effect that is not life-threatening, but it may lead to the interruption of the treatment or may alter the patient’s quality of life [10].

One of the targets that significantly impacted treatment of hematological malignancies is BCL2 since its discovery by the team of Croce et al. [11,12,13,14]. Currently, the most used drug that targets BCL2 is venetoclax. Currently, this drug impacts a variety of hematological malignancies, most notably chronic lymphocytic leukemia (CLL) and acute myeloid leukemia (AML) [15,16]. The main side-effect arises from the anti-apoptotic role of BCL2. Because of this, venetoclax is linked to a high tumor lysis syndrome when used without caution. Nonetheless, this side-effect can be avoided in most cases through correct dose escalation, prior cytoreduction and renal protection [17].

The checkpoint blockade in cancer was awarded the 2018 Nobel prize in Medicine or Physiology [18]. It acts by releasing some of the regulatory effects that occur either between the T-cell and the antigen-presenting cell (APC) or between the T-cell and the malignant cell. Because T-cells are likely to recognize a more altered antigen repertoire as non-self, checkpoint blockers have a more significant effect in malignancies with a higher degree of genomic instability. Because of this, nivolumab and pembrolizumab were approved for use in any malignancy with genomic instability [19]. The main side-effect of this class of molecules is caused by their T-cell activation mechanism. Still, although it enhances the action of T-cells against malignant cells, it also increases the probability for T-cells to recognize self-structures as non-self, leading to autoimmune conditions and associated side-effects [20].

Immunomodulators, represented by thalidomide, lenalidomide and pomalidomide, act by altering the targets in the proteasomal degradation pathway, leading to an increased degradation of Ikaros family zinc family protein 1 (IKZF1) and IKZF3, which in turn reduces myelocytomatosis (MYC) transcription. Moreover, this class of molecules inhibits the overall proteasomal degradation pathway, leading to protein accumulation and thus being important for diseases which produce an increased number of proteins, as in the case of multiple myeloma [21,22]. Apart from the teratogenic effects that thalidomide is known for, the main side-effects that occur in patients treated with these compounds are pancytopenia and neuropathic pain. However, besides these two side-effects, the drugs are generally well tolerated [23,24].

Fms-like tyrosine kinase (FLT3) is an important target especially for FLT3-internal tandem duplication (ITD) AML, as this alteration is associated with a bad prognosis [25]. The first generation of FLT3 inhibitors is most frequently represented by midostaurin and the second by gilteritinib and quizartinib. The side-effect profile of this class is generally well tolerated with increased probability of developing cytopenias when combined with venetoclax [25,26,27].

BTK inhibitors are very efficient for the treatment of CLL and have few side-effects, the downside being a life-long therapy. The side-effect profile is generally well tolerated, so extended use of these drugs does not lead to significant discomfort for most patients. Nonetheless, ibrutinib, the most commonly used BTK inhibitor, causes blood thinning and arrhythmias and must be stopped 3 days before and after a minor surgical intervention, as well as 7 days before and after a major surgical intervention. Considering that CLL develops over a long period of time, these stops are generally not of concern [28].

PI3K inhibitors have been included in several clinical trials for lymphoid diseases. Nonetheless, an important percentage of them were not validated clinically because of the high degree of side-effects that can occur. The most frequently used compound in this case is idelalisib, which induces severe diarrhea in many patients. Because of this toxicity, clinical trials for this drug were generally conducted as monotherapy, as when used in combined therapy they revealed an unacceptable side-effect profile. Nonetheless, these drugs have been shown to be very active in lymphoid diseases, thus having potential clinical use if the side-effect profile can be better managed [29]. An important drug that revolutionized symptom management in primary myelofibrosis is ruxolitinib [30,31,32]. This drug inhibits both JAK1 and JAK2, thus reducing proinflammatory cytokine levels and, because of this, the size of the spleen and severity constitutional symptoms. The side-effects that occur in this case are related to the role of JAK2 as an important transducer in myeloid differentiation and of JAK1 in other cytokine signaling pathways [33,34,35]. The first is responsible for inducing pancytopenia to the point that ruxolitinib has to be reduced or discontinued, and the latter transducer could potentially be responsible for the extreme weight gain that these patients observe under treatment [36,37].

## 3. CAR-T Cells and Bispecific Antibodies

A novel category of therapies that must be considered is represented by CAR-T cells. These are T-cells that are engineered to present a receptor for a specific antigen that, when encountering the antigen, activate the engineered T-cell. A common effect among this class of therapeutics is cytokine release syndrome (CRS) that occurs between 60% and 80% of treated patients [38,39,40,41]. CRS is characterized by a massive release of inflammatory cytokines and subsequent multiorgan dysfunction. Additionally, because of these cytokines, neurological and pulmonary symptoms can occur separately from CRS. In addition to these class effects, side-effects can also be in accordance to the target of the CAR-T cell, most notably anti-CD19, also requiring intravenous immunoglobulins [42].

A similar class of therapeutics considering the mechanism of action is represented by bispecific antibodies. In these cases, one arm of these molecules binds CD3 on T-cells and the other arm binds to the target antigen, generally represented by CD19 on the malignant clone. Of note, these also present CRS, but in a smaller proportion compared to CAR-T cells [43].

To offer an overview of the side-effects of the therapies that are frequently used, we classified them according to the FDA approval status and on the systems affected (Figure 2, Figure 3, Figure 4, Figure 5, Figure 6, Figure 7 and Figure 8), and we offer a summary of their molecular mechanisms that might explain the observed side-effects (Table 1).

## 4. Conclusions

In the current concise review, we offered an overview of the most common therapies that are used in hematology with the main interest that other specialties would find this of use when collaborating with a hematologist. Nonetheless, this should only serve as a primer for understanding the hematologic patient and should not substitute discussions with a hematologist or further consulting the specific literature.

## Figures and Tables

**Figure 1 jcm-09-02903-f001:**
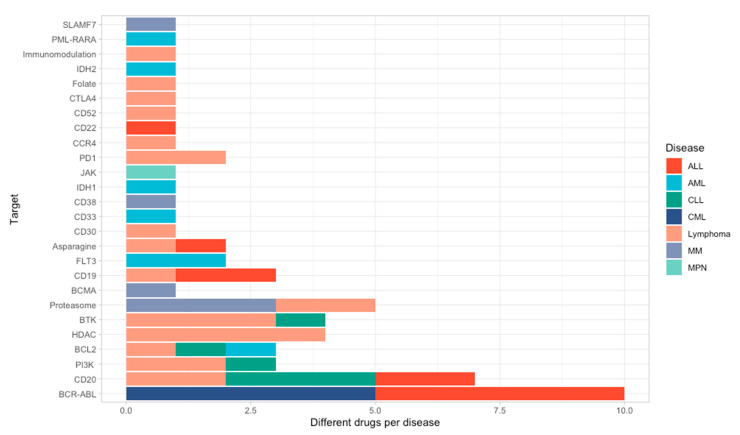
Targets of different FDA-approved and non-approved drugs across different hematological disorders; Legend: acute lymphoblastic leukemia (ALL); acute myeloid leukemia (AML), chronic lymphocytic leukemia (CLL), chronic myeloid leukemia (CML), multiple myeloma (MM), myeloproliferative neoplasm (MPN).

**Figure 2 jcm-09-02903-f002:**
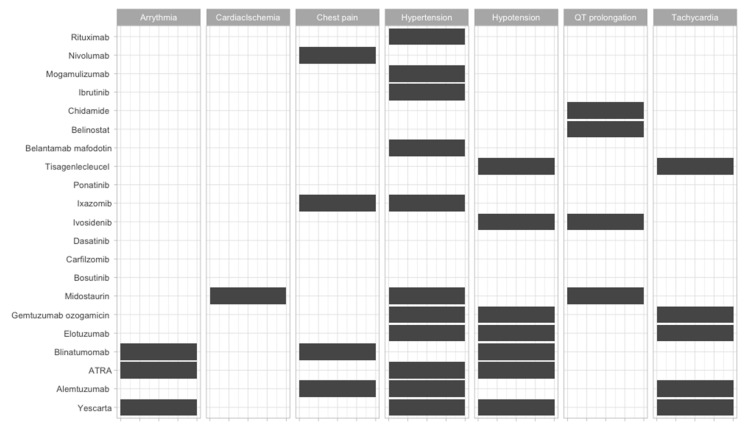
Cardiac side-effects of Food and Drug Administration (FDA)-approved and non-approved drugs for hematological disorders.

**Figure 3 jcm-09-02903-f003:**
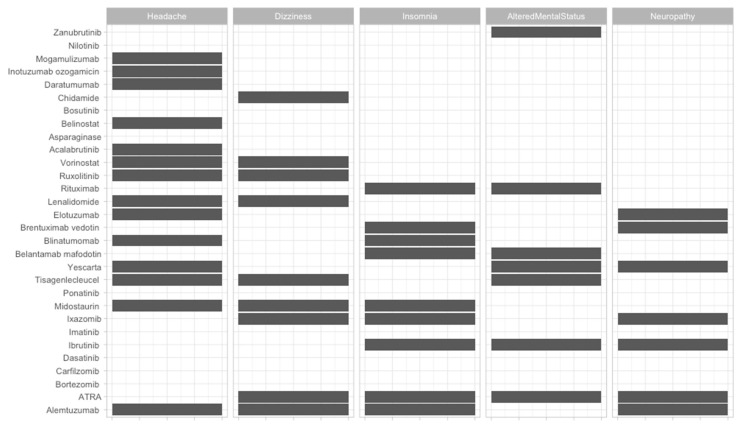
Neurologic side-effects of FDA-approved and non-approved drugs for hematological disorders.

**Figure 4 jcm-09-02903-f004:**
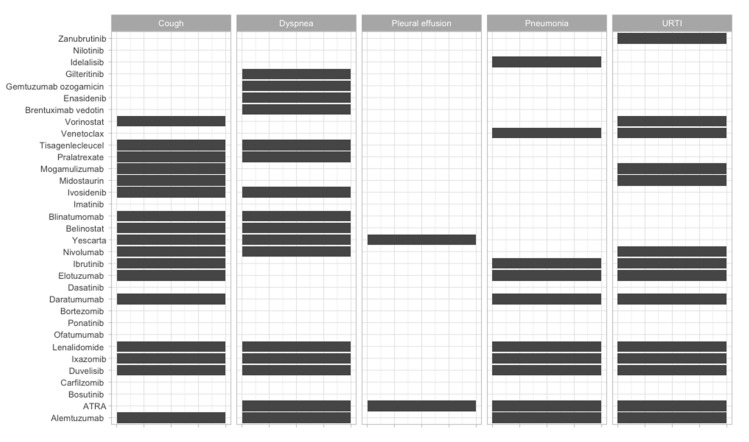
Respiratory side-effects of FDA-approved and non-approved drugs for hematological disorders. URTI = upper respiratory tract infection.

**Figure 5 jcm-09-02903-f005:**
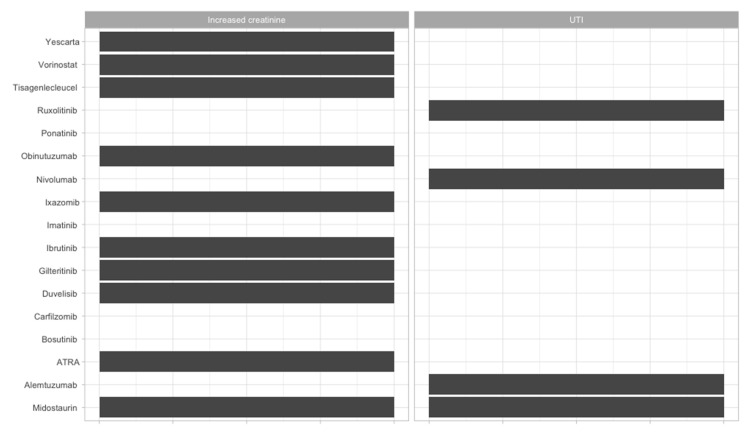
Renourinary side-effects of FDA-approved and non-approved drugs for hematological disorders. UTI = urinary tract infection.

**Figure 6 jcm-09-02903-f006:**
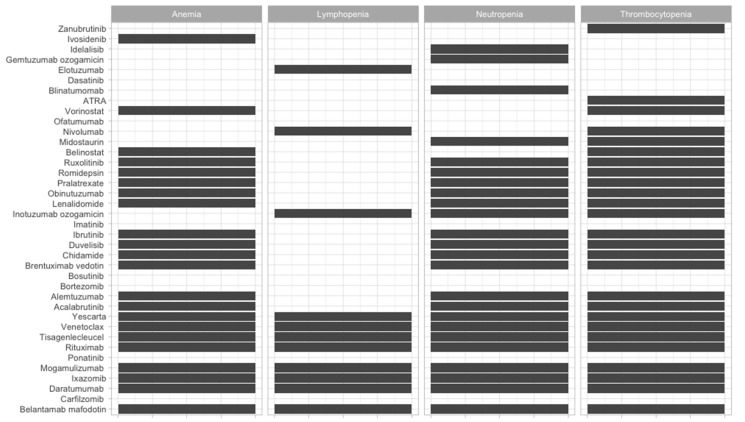
Hematologic side-effects of FDA-approved and non-approved drugs for hematological disorders.

**Figure 7 jcm-09-02903-f007:**
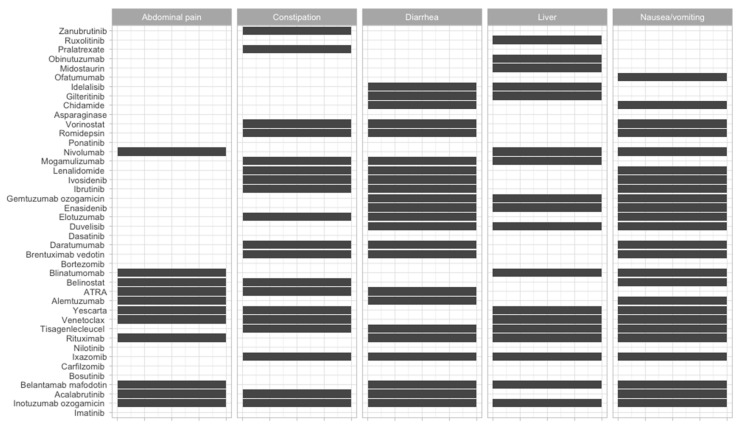
Gastrointestinal side-effects of FDA-approved and non-approved drugs for hematological disorders.

**Figure 8 jcm-09-02903-f008:**
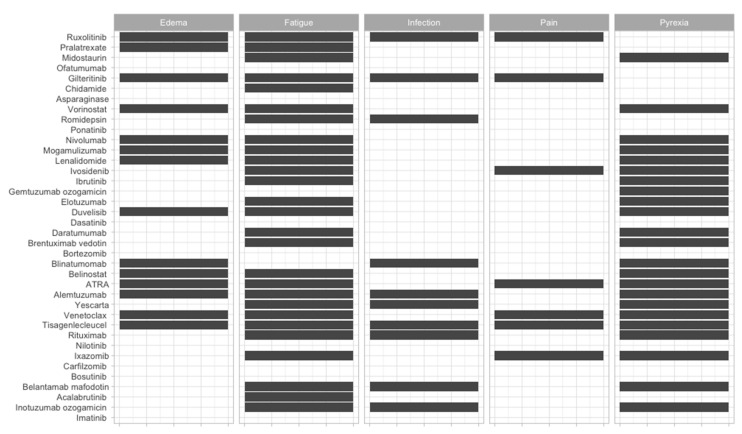
Systemic side-effects of FDA-approved and non-approved drugs for hematological disorders.

**Table 1 jcm-09-02903-t001:** Molecular/biological mechanisms behind the observed side-effects of the most common therapeutic strategies applied in hematological malignancies.

Therapeutic Strategy	Type of Targeted Cell	Type of Hematological Malignancy	Mechanism behind Observed Side-Effect	Ref.
BCR-ABL inhibitor	Malignant myeloid cell, malignant lymphoid cells	Chronic myeloid leukemia, acute lymphocytic leukemia	⮚ non-specific targeting of other kinases ⮚ increase in endoplasmic reticulum stress response ⮚ Jun N-terminal kinases non-specific inhibition, leading to mitochondrial disfunction ⮚ lowering the available levels of ATP ⮚ targeting of c-Kit leading to skin hypopigmentation	[44]
anti-CD20 targets	B cell (malignant, as well as normal B cells)	B-cell non-Hodgkin’s lymphoma, B-cell leukemias	⮚ decrease in total B cell count ⮚ depletion of normal B cells ⮚ indirect effects on dendritic cells and T cells	[45]
anti-CD30	Reed–Sternberg cells in Hodgkin lymphoma, LCL cells, other subtypes of NHL	Primary cutaneous anaplastic large-cell lymphoma, B-cell lymphoma, multiple myeloma, adult T-cell leukemia/lymphoma, mast cell malignancies	⮚ CD30 has a very low expression in normal tissue in comparison with lymphoma cells ⮚ expressed only in the rim of the follicular centers, proliferating germinal centers, and thymic medulla ⮚ expressed by activated CD8+/CD4+ T cells and activated B cells	[46]
BCL2 inhibitors	chronic lymphocytic leukemia (CLL) and acute myeloid leukemia (AML)	Chronic lymphocytic leukemia (CLL) and acute myeloid leukemia (AML)	⮚ BCL-2 is a major anti-apoptotic factor in the mitochondria that suppresses Bax and Bak molecules and causes the release of cytochrome C ⮚ in the endoplasmic reticulum, BCL-2 depletes the levels of Ca^2+^ ⮚ BCL-2 disrupts Ca delivery between the endoplasmic reticulum and mitochondria ⮚ after BCL-2 inhibition, cells undergo apoptosis very fast, the tumor cells die more rapidly than the ability of local macrophage to engulf the cell debris (tumor lysis syndrome) ⮚ tumor debris causes mineral imbalance in the blood circulation	[16,17,47]
immune checkpoint inhibitors (anti-CTLA4 and anti-PD1)	T-cells and APCs, or T-cells and malignant cells	malignancies with high genomic instability	⮚ physiological immunosuppression is stopped throughout the whole body ⮚ immune system begins to react more to the self-antigens (autoimmunity)	[48,49]
Immunomodulators (thalidomide, lenalidomide and pomalidomide,)	plasma cells	Multiple myeloma	⮚ binds to cereblon, a E3 ubiquitin ligase, that, together with DNA binding protein 1 (DDB1), Cullin-4A (CUL4A) and regulator of cullins 1 (ROC1), form the ubiquitin complex that ubiquitinates several proteins ⮚ IKZF1 and IKZF3 transcription factors, involved in B and T cell development are degraded ⮚ B and T cell development is impaired	[50,51]
anti-B-cell maturation antigen therapy	plasma cells	Multiple myeloma	⮚ specific for mature B cells, not found in hematopoietic stem cells ⮚ essential for the survival of plasma cells, but not for the overall survival of B-cells	[7,52,53]
PI3K inhibitors	B cells	Lymphoid diseases (especially of B-cell origin)	⮚ T regs are no longer capable of preventing auto-reactivity of the effector T-cells (auto-immune effects) ⮚ angiotensin II cannot function without PI3K activation (cardiovascular events) ⮚ PI3K inhibition slows calcium intake in vascular smooth muscle cells (cardiovascular events)	[29,54,55]
JAK1 and JAK2 inhibitors	megakaryocytes, but JAK1/JAK2 are ubiquitously expressed	Myelofibrosis, diffuse large B-cell lymphoma, and peripheral T-cell lymphoma.	⮚ JAK2 is involved in myeloid maturation, its down-regulation causes pancytopenia ⮚ JAK1 is involved in production of pro-inflammatory cytokines, its decrease, increases the susceptibility to infections	[56,57,58]
CAR T cells	depending on the engineered target	Wide applications in hematological malignancies, depending on the engineered target	⮚ CAR T cells are engineered to be activated and continuously proliferate when encountering neoantigens ⮚ CAR T cells form a positive feedback loop: the more tumor cells they encounter, the more CAR T cells will be activated and proliferate ⮚ CAR T cells cause rapid increase in pro-inflammatory cytokines	[39,40,41,42,53]

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
