# Peer review of "Overview of the Side-Effects of FDA- and/or EMA-Approved Targeted Therapies for the Treatment of Hematological Malignancies"

_jcm, 2020, doi:10.3390/jcm9092903_

Round 1
Reviewer 1 Report
Constantinescu and collegues reviewed side effects form drugs commonnly used to treat hematological diesases that could interest doctors that work in ICU, internal medicine and other specialities. The work is nice to read also for an hematologist and it is well written. Original figures and tables are usefull. The comprehensive purpose of the work obviously offer some grey spots and force authors at making semplifications however these semplifications do not impact on clarity nor on message.
Major observation:
I apreciate that the text is concise and give just some examples and that figures and tables has the merit to condensate an extensive ammount of knowledge. This ammount of knowledge needs for references.
I have some observation:
- Title: do not reflect text. You do not speak only of life-treating effects and include also mild effect. Figures and tables do not reflect severity of adverse event. Please modify title and text in a way in which clinicians can easly understand that not all the AEs from drugs are severe, life-treating or fatal (e.g. insomnia).
- page 1 line 34+36, ranged is repeated
- figure 1 has to be reviewed. Some data do not reflect the actual situation. For the complexity of the figure is it possible that I do not spot all the issue, I reccomend the author to review with great accurance.
- figure 1: BCR-ABL inhibitors was approved also for ALL
- figure 1: does PD1 account for anti PD1 or also CTLa4 and pdl1?
- figure 1: bcma monoclonal was approved by FDA
- figure 1: MEKi is not approved for AML
- figure 1: proteasome inhibitors does not account drugs in MM
- figure 1-2-3-4-5: FDA non approved drugs comprehend a wide cathegory of drugs, plrease restrict focus. It could be nice to have reference for each drug in figure.
- Different toxicities of BCR-ABL1i are extensivelly reported. Please cite some specific review.
- page 3 line 99, review the sentence
- figure 2: 2 different car-T cell was approved for lymphoma
- figure 3-4-5-6-7-8: are not carfilzomib and rituximab FDA approved? re-check all the drugs to allocate in the correct category. Please replace AG-120 and AG-221 bgb3111 and other drugs with the actual name.
- consider adding asparaginase
- Use acronyms: e.g. mental status is not an AEs, and it is clearly used in figure to shrink a longer sentence, URTI is not an universal acronym and it is not clarified.
- table 1: ther is an extra c on CD30 line (cells?), there is an extra line at the bottom, BCR-ABL target also lymphoid cell, it depends on translocation.
Author Response
Reviewer 1
Constantinescu and collegues reviewed side effects form drugs commonnly used to treat hematological diesases that could interest doctors that work in ICU, internal medicine and other specialities. The work is nice to read also for an hematologist and it is well written. Original figures and tables are usefull. The comprehensive purpose of the work obviously offer some grey spots and force authors at making semplifications however these semplifications do not impact on clarity nor on message.
Major observation:
I apreciate that the text is concise and give just some examples and that figures and tables has the merit to condensate an extensive ammount of knowledge. This ammount of knowledge needs for references.
Thank you for your feed-back. We have added more references.
I have some observation:
- Title: do not reflect text. You do not speak only of life-treating effects and include also mild effect. Figures and tables do not reflect severity of adverse event. Please modify title and text in a way in which clinicians can easly understand that not all the AEs from drugs are severe, life-treating or fatal (e.g. insomnia).
Thank you for your feed-back. We have modified the title to: “An Overview of the Side-Effects of FDA and/or EMA -Approved Targeted Therapies for The Treatment of Hematological Malignancies”. We removed life-threatening effects mentions from the text. We understand that a side-effect can have various severities, but, considering the approach of the current manuscript, to mention the severity for all the mentioned drugs would make the manuscript overly complex, which would make it too hard to read for a quick look-up.
- page 1 line 34+36, ranged is repeated
Thank you for your feed-back. We removed the first “ranged”.
- figure 1 has to be reviewed. Some data do not reflect the actual situation. For the complexity of the figure is it possible that I do not spot all the issue, I reccomend the author to review with great accurance.
- figure 1: BCR-ABL inhibitors was approved also for ALL
Thank you for your feed-back. We added this information.
- figure 1: does PD1 account for anti PD1 or also CTLa4 and pdl1?
Thank you for your feed-back. PD1 accounts only for antiPD1. CTLA4 is noted further right.
- figure 1: bcma monoclonal was approved by FDA
Thank you for your feed-back. We added this information.
- figure 1: MEKi is not approved for AML
Thank you for your feed-back. We removed MEKi from FDA approved.
- figure 1: proteasome inhibitors does not account drugs in MM
Thank you for you feed-back. We added proteasome inhibitors for MM.
- figure 1-2-3-4-5: FDA non approved drugs comprehend a wide cathegory of drugs, plrease restrict focus. It could be nice to have reference for each drug in figure.
Thank you for your feed-back. We removed the drugs that were not FDA approved from the figures.
- Different toxicities of BCR-ABL1i are extensivelly reported. Please cite some specific review.
Thank you for your feed-back we have added additional references for BCR-ABL1i.
- page 3 line 99, review the sentence
Thank you for your feed-back. We replaced the mentioned sentence with: “Currently, the most used drugs that targets BCL2 is represented by Venetoclax.”
- figure 2: 2 different car-T cell was approved for lymphoma
Thank you for your feed-back. Because we removed the non-FDA approved drugs, we decided to remove Figure 2 as it would have become redundant.
- figure 3-4-5-6-7-8: are not carfilzomib and rituximab FDA approved? re-check all the drugs to allocate in the correct category. Please replace AG-120 and AG-221 bgb3111 and other drugs with the actual name.
Thank you your feed-back. We made the necessary changes.
- consider adding asparaginase
Thank you for your feed-back. We added asparaginase.
- Use acronyms: e.g. mental status is not an AEs, and it is clearly used in figure to shrink a longer sentence, URTI is not an universal acronym and it is not clarified.
Thank you for your feed-back. We made the necessary corrections and specifications.
- table 1: ther is an extra c on CD30 line (cells?), there is an extra line at the bottom, BCR-ABL target also lymphoid cell, it depends on translocation.
Thank you for your feed-back. We removed the extra “c”. We added malignant lymphoid cells on the “type of targeted cells” column and “acute lymphocytic leukemia” on the “type of hematological malignancy” column from the BCR-ABL inhibitor line.

Reviewer 2 Report
Dear Authors, thank you for the opportunity to review your work. The manuscript presents an important aspect of chemotherapeutic drugs. The literature discussed and the presented is very intriguing and engaging. I would appreciate it if you can improve the quality of the figures so that the readers can appreciate the information presented.
Author Response
Reviewer 2
Dear Authors, thank you for the opportunity to review your work. The manuscript presents an important aspect of chemotherapeutic drugs. The literature discussed and the presented is very intriguing and engaging. I would appreciate it if you can improve the quality of the figures so that the readers can appreciate the information presented.
Thank you for your feed-back and your compliments. We have changed the format that the images are presented in so that they can be read more easily. Hopefully the resolution is fine now.

Reviewer 3 Report
This short review is well conducted and very explanatory.
I am very pleased with the style used, very clear.
The figures the authors are presenting are well done.
Author Response
Reviewer 3
This short review is well conducted and very explanatory.
I am very pleased with the style used, very clear.
The figures the authors are presenting are well done.
Thank you for your compliments.

Round 2
Reviewer 1 Report
Great work! I've no additional comments
Reviewer 2 Report
Thank you, dear authors, for changing the style of the manuscript.